# Separation and Determination of Chemopreventive Phytochemicals of Flavonoids from *Brassicaceae* Plants

**DOI:** 10.3390/molecules26164734

**Published:** 2021-08-05

**Authors:** Sylwia Bajkacz, Magdalena Ligor, Irena Baranowska, Bogusław Buszewski

**Affiliations:** 1Department of Inorganic Chemistry, Analytical Chemistry and Electrochemistry, Faculty of Chemistry, Silesian University of Technology, 6 Krzywoustego Str., 44-100 Gliwice, Poland; sylwia.bajkacz@polsl.pl (S.B.); irena.baranowska@polsl.pl (I.B.); 2Department of Environmental Chemistry and Bioanalytics, Faculty of Chemistry, Nicolaus Copernicus University, 7 Gagarina Str., 87-100 Torun, Poland; 3Interdisciplinary Centre of Modern Technologies, Nicolaus Copernicus University, 4 Wileńska Str., 87-100 Torun, Poland

**Keywords:** cabbage family, flavonoids, phenolic acids, glycosides

## Abstract

The main aim of this study was to develop a method for the isolation and determination of polyphenols—in particular, flavonoids present in various morphological parts of plants belonging to the cabbage family (*Brassicaceae*). Therefore, a procedure consisting of maceration, acid hydrolysis and measurement of the total antioxidant capacity of plant extracts (using DPPH assay) was conducted. Qualitative analysis was performed employing thin-layer chromatography (TLC), which was presented to be a suitable methodology for the separation and determination of chemopreventive phytochemicals from plants belonging to the cabbage family. The study involved the analysis of 25 vegetal samples, including radish, broccoli, Brussels sprouts, kale, canola, kohlrabi, cabbage, Chinese cabbage, red cabbage, pak choi and cauliflower. In addition, selected flavonoids content in free form and bonded to glycosides was determined by using an RP-UHPLC-ESI-MS/MS method.

## 1. Introduction

Plants of the cabbage family (crucifers) are widespread around the world. The greatest variety of *Brassicaceae* species can be found in South-East Asia, the Mediterranean region and the western part of North America. These plants can adapt to hostile environmental conditions and can develop even at high altitudes. *Brassicaceae* play an important role in human nutrition. Almost any morphological part of the plant can be used for consumption for some species, such as roots and tubers, leaves, transformed stems and generative organs (e.g., inflorescence or seeds) [1]. *Brassicaceae* plants are used as food and as a source of oil, fodder and medicinal herbs. Quite a few decorative species can be easily found in flowerbeds. There are cabbage cultivars suitable for long-term cold storage, and these can be sold up to 8 months after their harvest. Different cabbage varieties can be fermented or pickled and can be eaten raw, blanched, boiled, fried and stewed with other seasonings [2,3]. Due to suitable weather and soils, *Brassicaceae* are a very popular crop in Poland [4,5].

*Brassicaceae* are a source of many vitamins (B, C, E and K), biotin, fiber and selen [6]. These plants also contain phenolic compounds, phytosterols and carotenoids, with glucosinolates being the substances most characteristic of this plant family [7,8,9,10,11]. What is notable is the presence of organosulfur sulforaphane, a representative of isothiocyanates, resulting from the decomposition of glucosinolates [12]. *Brassicaceae* demonstrates highly antioxidative properties [13,14,15,16]. The specific contents of antioxidants differ depending on the plant species, its maturity, harvest time, cultivation conditions, soil quality, storage and transport conditions, and the thermal processing used during the preparation of cabbage dishes [17,18,19,20]. Depending on the species and the plant part intended for further use, the amount of available raw material can vary. For example, analysis of white cabbage leaves revealed over 90% of water content [21]. Other main components include carbohydrates, proteins, fats and other biologically active substances, as well as macro- and microelements (Figure 1) [21].

Due to the presence of biologically active compounds, the consumption of different cabbage species is generally beneficial for health [22]. Glucosinolates present in *Brassicaceae* plants play a significant role in cancer chemoprevention. Inhibition of carcinogenic processes by consuming plant-derived substances is one of the strategies proposed by preventive medicine. However, natural plant substances can only block and prevent the early stages of carcinogenesis, while in the later stages their influence is much weaker. In vitro and in vivo studies have demonstrated the chemopreventive properties of isothiocyanates and indoles, such as sulforaphane, phenyl ethyl isothiocyanate, indole-3-carbinol and the product of its condensation—3,3′-diindolylmethane [22]. The chemopreventive potential of the cabbage family is also linked to the presence of antioxidants. When included in the daily food intake, vitamins with antioxidative properties and polyphenols support the natural mechanisms of organism defense against free radicals [23,24,25,26]. These plants’ biological and health-promoting properties include their anti-bacterial, anti-inflammatory, anti-allergenic cytotoxic and nematicidal activities [26,27].

The *Brassicaceae* vegetables, which include cabbage, broccoli, cauliflower, Brussels sprouts, kohlrabi and kale, contain natural goitrogens [28]. The harmful activity of goitrogens involves binding to iodine, disrupting the proper synthesis of thyroid hormones [28]. This condition may lead to the excessive growth of the thyroid gland, resulting in goiter formation and the development of hypothyroidism. Such side effects of goitrogens intake can appear in individuals with iodine deficiency or when the diet delivers large amounts of these substances (e.g., due to excessive consumption of raw and processed cabbage) [28,29].

As in the case of other plants, the effective isolation of biologically active compounds from the cabbage family can be achieved by using organic solvents. Sonication is often used to improve extraction efficiency. Extracts can be analyzed by techniques such as thin-layer chromatography (TLC) and high-performance liquid chromatography (HPLC), combined with mass spectrometry (MS) or spectrophotometry [30,31,32,33,34,35].

This study was focused on the isolation and determination of polyphenols, in particular, flavonoids in *Brassicaceae* plants. Unlike glucosinolates, the incidence and content of polyphenols in cabbage family plants remain unknown due to the inter-species variability. In presented studies, antioxidant activity and total phenolics contents in plants extracts were discussed. Moreover, the thin layer chromatography and reversed-phase ultra-high-performance liquid chromatography coupled with a tandem mass spectrometry (PRP-UHPLC-MS/MS) methods were applied for analysis of selected polyphenols (flavonoids and phenolic acids) in 13 different *Brassicaceae* plants (radish, broccoli, brussels, kale, canola, conehead cabbage, green cabbage, chinese cabbage, pakchoi cabbage, red cabbage, Italian cabbage, kohlrabi and cauliflower).

## 2. Results

### 2.1. Extraction Yield

The extraction yield (*EY*) has been evaluated by the measurements of the dry extract residue mass after evaporation of organic solvent. *EY* was calculated for the each sample as a quotient the mass of dry extract and the mass of dry sample of plant material. Obtained results for each sample are presented in Table 1.

### 2.2. Hydrolysis of Extracts

The qualitative analysis of polyphenols in obtained extracts allowed for the evaluation of hydrolysis method efficiency. The evaluation criteria considered the number of identified polyphenols, repeatability and lack of interfering substances. In the present work, the hydrolysis method (Figure 2) has been applied to identify and determine flavonoid obtained through the maceration of *Brassicaceae* plants. Our investigation aimed to improve the hydrolysis and clean-up methods commonly used for simultaneous determination of flavonoid in various morphological parts of plants.

First of all, to examine the composition and characteristics of obtained extracts after acid hydrolysis, preliminary chromatographic analyses were performed using TLC (Figure 3).

The obtained results confirm the appropriateness of the used methodology, particularly regarding the evaluation of leaf extracts, once several compounds were visualized for TLC analysis of samples No 4, 8, 11, 15, 19 and 24. For these samples, the presence of a larger number of biologically active compounds can be assumed. The presence of compounds such as flavone,7-hydroxy flavone, rutin, quercetin, naringin, naringenin, esculin, esculetin, biochanin A, gallic acid, salicylic acid, cumaric acid, chlorogenic acid and sinapinic acid was confirmed by TLC analysis of a mixture containing standards of flavonoids and polyphenolic acids.

### 2.3. Antioxidant Activity and Total Phenolics Contents in Plants Extracts

Spectrophotometric methods were used to characterize and evaluate the chemopreventive properties of the investigated extracts. The antioxidant activity of methanolic extracts from plants belonging to the cabbage family was studied. Inhibition times for the studied extracts were 15 min. After this period, absorbance was measured (λ = 517 nm). The results concerning radical scavenging activity (RSA) are presented in Table 2. The total antioxidant activity of the plant extracts was calculated using the following Equation (1):(1)$$RSA=\frac{A_{DPPH}-A}{A_{DPPH}}\cdot 100\%$$
where *A*—absorbance of the mixture of DPPH^●^ with plant extract after inhibition time; *A_DPPH_*—absorbance of pure DPPH^●^ solution; and *RSA*—radical scavenging activity.

The other purpose of these assays was to measure the total non-enzymatic antioxidant capacity (TAC) of samples, by the evaluation of their ability to counteract oxidative stress-induced damage in cells. TAC was used to provide insights into the development and treatment of oxidative-stress related disorders. Since the applied assay kit gives antioxidant capacity in Trolox equivalents, TAC was expressed in terms of Trolox equivalent antioxidant capacity (TEAC). As it is known, Trolox is a water-soluble vitamin E analog, therefore serving as an antioxidant standard. The results regarding %RSA, TEAC and total phenolic content are presented in Table 2.

Based on obtained results, all analyzed extracts showed total phenolic acid content over 4 mg/g with the exception of canola roots, kohlrabi spots and kohlrabi edible spot (which had slightly lower phenolic acids content). The amounts of total phenolic are very similar to those found by Šamec et al. (2018) in the case of broccoli [36], kale and cabbage, by Fratinni et al. (2014) for broccoli [37] and by Heimler et al. (2006) for cabbage, brussels and cauliflower [38]. Comparisons of antioxidant capacity of the *Brassicaceae* have been studied by number of authors [10]. Significant variability was noted among different cultivars as well as among genotypes of the same cultivar from different geographical origins, which was in agreement with our findings (e.g., different types of cabbage).

In our experiments, radish sprouts showed the highest amount of total phenolic content and antioxidant capacity expressed both as RSA and TEAC. Total phenolic content of radish sprouts was around 50% higher than those, obtained for radish roots, spots and leaves. Considering broccoli, we also found that the examined sprouts contain significantly higher amounts of total phenolic than flowers. In parallel with the high content of total phenolic compounds, broccoli sprouts showed the highest DPPH radical scavenging activity and TEAC. Our data also confirmed that, regarding sprouts, other *Brassicaceae* species (e.g., radish, kale) could also be good candidates as a source of health promoting compounds, and surely they deserve more scientific attention. Interestingly, broccoli is, so far, the most commonly studied *Brassicaceae* regarding health benefits recognized as a vegetable with a high antioxidant capacity [10,23,24,25].

It can be concluded that plants that belong to the cabbage family contain substantial amounts of biologically active compounds—in particular, sprouts appear to be a rich source of antioxidants (especially polyphenols). These compounds have chemopreventive properties and affect the nutritional value of *Brassicaceae* plants.

### 2.4. RP-ESI-UHPLC-MS/MS Method Validation

The linearity range, correlation coefficients (R^2^), LOD and LOQ of target analytes are listed in Table 3. The R^2^ values were all higher than 0.991, revealing good linearity for the concentration range studied. LOD and LOQ were in the ranges of 0.13–6.67 ng/g and 0.4–20 ng/g, respectively, demonstrating the high sensitivity of the developed method.

The intra- and inter-day accuracy and precision at two QC levels were determined (Table 3). The relative error (RE) values for intra-day assays ranged from −6.42% to 5.87%, and the coefficient of variation (CV) was <6.71%. The RE values for inter-day assays ranged from −8.52% to 7.85%, and the CV values were <8.69%. These results indicate that the present method accurately and reproducibly measures each analyte. Moreover, the matrix effect is presented in Table 3. The ME values were between −8.73% and 6.90%, indicating that the matrix effect on the response of target analyte was not obvious under the developed conditions. The above validation results indicated that the proposed method could be used to simultaneously determine all the selected bioactive compounds in plants from the *Brassicaceae* family with high precision, sensitivity and accuracy.

### 2.5. Determination of Phenolic Acids and Flavonoids in Brassicaceae Plants Extracts

According to RP-ESI-UHPLC-MS/MS experiments, we evaluated in this study the main phytochemicals associated with health benefits in *Brassicaceae* species, including flavonoids and phenolic acids (Figure 4, Appendix A). Extracts with high amounts of polyphenols (over calibration curve) were diluted before analysis. In all studied samples, the highest amount of phenolic acids was observed compared to flavonoids. Four phenolic acids (3,4-DHBA, 4-HBA, 3,4-HPPA and *p*-COA) were determined in each plant, while the most abundant phenolic acid was 4-HBA. Significantly higher content of phenolic acids, especially 4-HBA, was found in radish and broccoli followed by commonly studied cauliflower. The concentration of phenolic acids was higher in radish roots (1.32–4054 ng/g) than in spots and leaves (0.78–1477 ng/g and 1.63–1663 ng/g). As regards polyphenol characterization, as an example, Figure 5 reports the chromatographic profiles of the extracts of (a) radish sprouts, (b) radish roots, (c) radish spots and (d) radish leaves. However, in the case of canola, we observed another dependence, and higher amount in leaves (2.21–833 ng/g) and spots (1.44–924 ng/g) than in roots (0.5–527 ng/g). When exploring the phenolic acids present in canola, 3,4-HPPA was the main compound in roots (527 ng/g), while *p*-COA was most present in the spots (924 ng/g) and leaves (833 ng/g). Extracts of cauliflower head showed the highest concentrations of the determined phenolic acids (1.21–1306 ng/g) compared to midvein (0.4–169 ng/g) and leaves (0.44–270 ng/g) (Figure 6). Among the analyzed cabbage extracts, the highest amounts of phenolic acids were determined in leaves of red cabbage (0.40–1167 ng/g), while 3,4-DHBA was the main compound present at a higher concentration level (1167 ng/g). Three phenolic acids (*α*-HHA, HA and HVA) were not detected or quantified in the whole extracts of studied plants. The data obtained for CA, FA, DOPAC and 3-HPA mostly indicated lower concentrations in comparison with other acids; moreover, these compounds were determined only in a few extracts. The concentration of 3-HBA and *p*-COA was similar to values previously quoted in the literature for broccoli, cabbage and cauliflower [39]. For the other brassica vegetables (brussels radish, kohlrabi), no comparative data are available as earlier studies of these vegetables focused on determining total phenolic content rather than concentrations of the individual phenolic acids [10].

Considering flavonoids, we found that all examined plants contain lower amounts of flavonoids than phenolic acids ones. Among the studied flavonoids, LQG, LIQ, FOR, FIS, NRI, GLB, NARG, PIN, NHSD, HSD and HST were not quantified in the analyzed plant extracts. These results seem to be justified, taking into account the main sources of the above-mentioned compounds. For example, GLB, LQG and LIQ are chemical compounds that are found in the root extract of licorice [40,41,42]; FIS can be found in many fruits and vegetables, such as strawberries, apples, onions and cucumbers [43]; PIN is found in honey [44]; and FOR can be found in found in red clover [45]. On the other hand, HSD and NARG are found in citrus fruits, and upon ingestion they release their aglycones, HST and NAR [46]. The compound with the highest concentration was RUT determined in radish leaves (178 ng/g). The most abundant flavonoids were QUE and NAR; however, their contents were very low, 0.92–61.2 ng/g and 0.4–2.0 ng/g, respectively.

As a summary of the analysis of different plants from the *Brassicaceae* family, it may be stated that the qualitative and especially quantitative polyphenol profiles are significantly different. It may be supposed that the pharmacological activities of the studied plants are not equal.

## 3. Material and Methods

### 3.1. Analyzed Samples 

The selected samples are plants from the *Brassicaceae* family. The research included several species and types of plants, which are listed in Table 4. The raw plant material was classified according to the morphological part—flowers, leaves, stems and roots. The individual samples were dried at 40 °C (in the dark) and then grounded. The grounded raw materials were kept in glass containers in the dark, until further processing.

### 3.2. Extract Preparation (Maceration)

In Falcon tubes, 1.00 g of plant material and 20 mL of 96% ethanol were combined. The contents were submitted to sonication 5 min. Next, the tubes were removed from the ultrasound bath and maceration was conducted for 24 h. The proposed method of extraction has been developed in our own laboratory. To isolate the extract from the solid residue, the sample was sifted through filter paper, using a Büchner funnel. The extracts were collected in Falcon tubes and dried under a nitrogen stream. To calculate the extraction effectiveness, the dry mass of the extraction residue was determined using an analytical balance. Next, extracts were dissolved in 2 mL of methanol, filtered through PTFE syringe filters (13 mm diameter, 0.22 μm pore size) and collected into vials.

### 3.3. Hydrolysis of Plant Extracts

Once the compounds of interest were present in the studied extracts in glycoside form, it was necessary to introduce an acid hydrolysis step to obtain free aglycones. The proposed procedure of hydrolysis was developed in our laboratory and is based on an experience in this field. To maintain a proper temperature, the hydrolysis was conducted in a thermostat (Eppendorf^®^ Thermomixer Comfort, Hamburg, Germany). The schematics for the procedure are presented in Figure 7. Prior to analysis, methanolic extracts were filtered through PTFE syringe filters (13 mm diameter, 0.22 μm pore size).

### 3.4. Chemicals and Reagents for Preliminary Experiments

Standards of flavonoids and phenolic acids (purity 99%) (chrysin (CHS) (used as an internal standard; IS), rutin (RUT), hesperetin (HST), quercetin (QUE), naringenin (NAR), naringin (NARG), narirutin (NRT), hesperidin (HSD), neohesperidin (NHSD), pinocembrin (PIN), taxifolin (TAX), fisetin (FIS), glabridin (GLB), eriocitrin (ERC), eriodictyol (ERI), formononetin (FOR), liquiritin (LIQ), liquiritigenin (LQG), 3-hydroxybenzoic acid (3-HBA), benzoic acid (BA), caffeic acid (CA), 3,4-dihydroxybenzoic acid (3,4-DHBA), hippuric acid (HA), α-hydroxyhippuric acid (α-HHA), 3-(4-hydroxyphenyl)propionic acid (3,4-HPPA), 4-hydroxybenzoic acid (4-HBA), 3,4-dihydroxy-phenylacetic acid (DOPAC), 3-hydroxyphenylacetic acid (3-HPA), *p*-coumaric acid (*p*-COA), ferulic acid (FA) and 4-hydroxy-3-methoxyphenylacetic acid (HVA)) were purchased from Roth (Karlsruhe, Germany) and Sigma Aldrich (St. Louis, MO, USA). Stock solutions were prepared by dissolving 1 mg the solid compounds in 1 mL of methanol obtaining concentrations of 1.0 mg/mL. Working solutions were prepared by appropriate dilution of stock solutions with methanol. Calibration standards (CSs) were prepared at ten levels (0.4; 4.0; 10; 30; 60; 100; 200; 300; 400; 500 ng/g), ranging from 0.4 to 500 ng/g, by dilution of the polyphenol working solutions with plant extracts. Quality control (QC) samples were at two concentration levels of polyphenols: 10 ng/g (low quality control, LQC) and 300 ng/g (high quality control, HQC). Methanol, petroleum ether, cyclohexane, ethyl acetate and acetone were purchased from J.T. Baker (Deventer, Holland). Water, formic acid, methanol and acetonitrile for LC–MS from Merck (Darmstadt, Germany) were used as a mobile phase. Hydrochloric acid (36–38%) was purchased from Avantor Performance Materials S.A. (Gliwice, Poland). Synthetic free-radical 2,2-diphenyl-1-picrylhydrazyl (DPPH^●^) was purchased from Sigma Aldrich (St. Louis, MO, USA). Water was obtained using MilliQ RG apparatus by Millipore Intertech (Bedford, MA, USA).

### 3.5. Apparatus

Analysis was performed using an HPTLC (high-performance thin layer chromatography) system from CAMAG (Muttenz, Switzerland), equipped with a Linomat V Applicator, Visualizer and VisionCATS data processor (version 2.0). TLC analysis of standards and plant extracts was performed in a DS-L horizontal chamber obtained from Chromdes (Lublin, Poland). TLC plates with silica gel on aluminum foil and plastic background Kieselgel 60 F_254_ (Merck, Darmstadt, Germany) were also used. The plates were visualized under UV light (λ = 254 nm and λ = 366 nm). The spectrophotometric measurements were performed by usage of UV-Vis spectrophotometer Helios Gamma (Thermo Fisher Scientific, Waltham, MA, USA) and spectrophotometric multiwell plate reader Varioscan (Thermo Scientific, Waltham, MA, USA). This apparatus was used for qualitative analysis of standards solutions and plant extracts.

The flavonoids and phenolic acids were analyzed using a Dionex UHPLC system (Dionex Corporation, Sunnyvale, CA, USA) equipped with an UltiMate 3000 RS (Rapid Separation) pump, an UltiMate 3000 thermostatted column compartment and an UltiMate 3000 autosampler. Dionex Chromeleon TM 6.8 software was used to control the UHPLC system. The UHPLC system was coupled with an AB Sciex Q-Trap^®^ 4000 mass spectrometer (Applied Biosystems/MDS SCIEX, Foster City, CA, USA). The system was controlled using the Analyst 1.5.1 software.

### 3.6. Antioxidant Activity

Evaluation of the antioxidant activity of the crude methanolic extracts was performed. For the preparation of DPPH solution, 2 mg of the pure substance was dissolved in 100 mL of methanol. Next, 100 μL of the methanolic extracts was added to 3 mL of DPPH solution, and this mixture was kept in the dark for 30 min. After this procedure, the absorbance of the mixture was measured using a spectrophotometer at wavelength λ = 517 nm. For control experiments, the absorbance of pure DPPH solution was also measured, using methanol instead of plant extract.

Briefly, in the course of this assay, Cu^2+^ ions are converted into Cu^+^ by both proteins and small antioxidant molecules. Then, the “protein mask” is required to prevent ion reduction by proteins. Next, the reduced Cu^+^ ions are chelated with a colorimetric probe. The absorbance of each sample was recorded at λ = 570 nm, being proportional to the total antioxidant capacity. The mentioned kit gives the antioxidant capacity in Trolox equivalents (ranging from 4–20 mM). Samples were analyzed using a multiwell plate reader.

### 3.7. Determination of Total Phenolics Contents in Plants Extracts

The total phenolics content was measured spectrophotometrically at λ = 760 nm, using the Folin–Ciocalteu reagent. This reagent is a mixture of phosphomolybdate and phosphotungstate, and it is widely used for the colorimetric assay of polyphenols. Standards of gallic acid with concentrations ranging from 0.1 to 1.3 mg/mL were also measured. The results were expressed as gallic acid equivalents (GAE) per gram of dry matter.

### 3.8. TLC Analysis

A mixture consisting of petroleum ether: cyclohexane: ethyl acetate: acetone: methanol (60:16:10:10:4; *v/v*) was employed as the mobile phase. The chromatograms on silica gel plates were processed for 30 min. The plates (10 × 20 cm) were covered with methanol extracts of mentioned plants. The mobile phase traveled a distance of 8.0 cm. The plates were dried at 40 °C, and images were captured employing a Visualizer under a UV lamp. The detection was performed at the wavelengths λ = 254 nm and 365 nm.

### 3.9. RP-UHPLC-ESI-MS/MS Method to Determine Selected Flavonoids

Thirty selected flavonoids and phenolic acids (rutin (RUT), hesperetin (HST), quercetin (QUE), naringenin (NAR), naringin (NARG), narirutin (NRI), hesperidin (HSD), neohesperidin (NHSD), pinocembrin (PIN), taxifolin (TAX), fisetin (FIS), glabridin (GLB), eriocitrin (ERC), eriodictyol (ERI), formononetin (FOR), liquiritin (LIQ), liquiritigenin (LQG), 3 hydroxybenzoic acid (3-HBA), benzoic acid (BA), caffeic acid (CA), 3,4-dihydroxybenzoic acid (3,4-DHBA), hippuric acid (HA), α-hydroxyhippuric acid (α-HHA), 3-(4-hydroxyphenyl) propionic acid (3,4-HPPA), 4-hydroxybenzoic acid (4-HBA), 3,4-dihydroxy-phenylacetic acid (DOPAC), 3-hydroxyphenylacetic acid (3-HPA), *p*-coumaric acid (*p*-COA), ferulic acid (FA), 4-hydroxy-3-methoxyphenylacetic acid (HVA) and chrysin (CHS) (used as an internal standard; IS)) were analyzed in plants extracts using RP-UHPLC-ESI-MS/MS method. The applied method was described previously by Bajkacz et al. (2018) [47]. Separation was performed using a Zorbax Eclipse XDB-C18 column (50 × 2.1 mm, 1.8 μm, Agilent Technologies, Santa Clara, CA, USA). As a mobile phase, 0.1% *v/v* formic acid in water and acetonitrile in gradient elution mode was applied. The gradient program used was as follows: (1) mobile phase A/B was set to 95%/5% at 0 min; (2) a linear gradient was dropped to 40%/60% A/B in 8.0 min; (3) mobile phase A/B was ramped to 95%/5% again in 0.1 min; and (4) from 8.1 to 10 min, mobile phase A/B was maintained at 95%/5%. The mobile phase flow rate was 0.5 mL/min, the injection volume was 5 μL and the column temperature was maintained at 30 °C. The total run time was 10 min.

Electrospray ionization (ESI) in the negative ionization mode was used for detection of polyphenols. Analysis was performed in the multiple response monitoring (MRM) mode. The optimized MS parameters for the selected MRM transitions for each compound are according to the data presented in our previous work [47]. The most intense transitions were used for quantification, and the other transitions were used for confirmation (Appendix A). The following settings were also applied to the turbo ion spray source: capillary voltage (IS), −4500 V; temperature (TEM), 500 °C; nebulizer gas (GS1), 60 psi; turbo-gas (GS2), 50 psi; curtain gas (CUR), 20 psi; and collision activated dissociation gas (CAD), 4 psi.

### 3.10. Methods Validation

In order to check the correctness of the applied method, validation parameters such as linearity, the limit of detection (*LOD*), limit of quantification (*LOQ*), precision, accuracy, the matrix effect and extraction efficiency were determined [48].

The linearity of the calibration curve was evaluated by analyzing polyphenol standard solutions in plant extract at ten concentrations ranging from 0.4 to 500 ng/g. Each concentration level was prepared with three replicates. The calibration curve was prepared by determining the best fit of the peak area ratio (peak area of analyte/internal standard) vs. concentration using a regression weighted by a factor of 1/x^2^. The limit of detection was determined by measuring the signal-to-noise ratio (S/N) and the limit of quantification using the Equation (2) [49]:(2)LOQ=3LOD

Precision, accuracy and matrix effect were determined at two concentration levels: low (LQC = 10 ng/g) and high (HQC = 300 ng/g). Intra-day precision within one day and inter-day precision in three continuous days were studied by observing three replicates of each target compound. In order to determine the precision, the coefficient of variation (%CV) was determined; the accuracy was assessed on the basis of the relative error (%RE). The matrix effect was calculated according to Equation (3), where *A_extract_* means the analyte area in the sample after extraction (sample No 21) and *A_standard_* means the analyte area in the standard solution [48]:(3)$$ME\left[\%\right]=\left(1-\frac{A_{extract}}{A_{standard}}\right)\cdot 100\%$$

The extraction capacity was determined after maceration using 20 mL of 96% ethanol and 1.00 g of dry and ground morphological parts of plants. After 24 h, obtained extracts were filtered into previously weighed plastic vials. After evaporation of the solvent, vials were weighed again. The extraction yield (*EY*) has been evaluated by the dry extract residue mass measurements after evaporation of organic solvent. Mentioned measurements been done by means of an analytical balance. The extraction efficiency was calculated according to Equation (4), where *M_extract_* means the mass of dry extract and *M_sample_* means the mass of dry sample of plant material:(4)$$EY\left[\%\right]=\left(\frac{M_{extract}}{M_{sample}}\right)\cdot 100\%$$

## 4. Conclusions

Obtained results showed that all twenty five analyzed plants from the *Brassicaceae* family contain phytochemicals with health-promoting benefits. A significantly high content of polyphenols and antioxidant activity were found in radish sprouts, followed by broccoli and kale sprouts. Both varieties have not, so far, been widely used for food as sprouts. Presented results showed that some kind of vegetables (canola, conehead cabbage and pak chio cabbage) contributed to the low antioxidant capacity.

The applied RP-ESI-UHPLC-MS/MS procedure was demonstrated to be effective for quantifying polyphenols in very complex matrixes such as broccoli, brussels, cabbage, cauliflower and kale. The main phenolic acids in the studied plants have been found to be 4-HBA, 3,4-HPPA and *p*-COA, which are the major polyphenols detected in the analyzed samples. Based on presented UHPLC-MS/MS results examined, radish, focusing on roots, deserves more scientific attention as a cheap source of phytochemicals with health-promoting benefits.

Summarizing, obtained results indicated that the plants from the *Brassicaceae* family evaluated may provide a potential source of dietary antioxidant, and therefore their consumption should be stimulated.

## Figures and Tables

**Figure 1 molecules-26-04734-f001:**
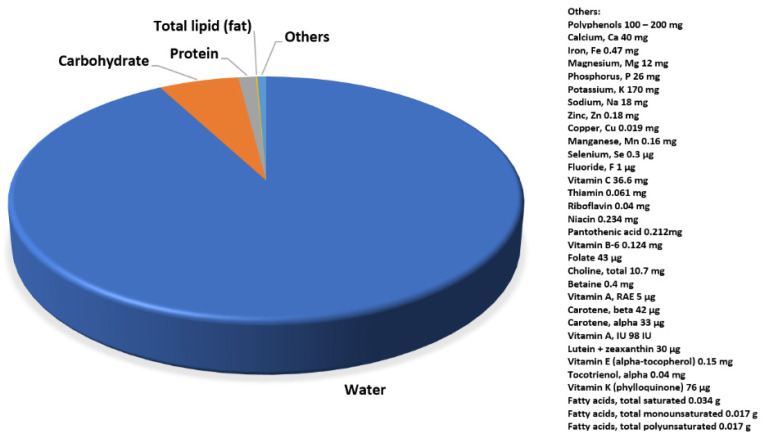
An example of raw cabbage leaves composition, partially according to [21].

**Figure 2 molecules-26-04734-f002:**
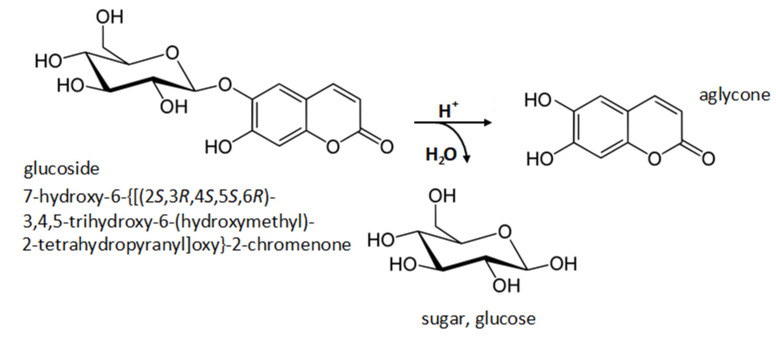
An example acid hydrolysis of aesculin.

**Figure 3 molecules-26-04734-f003:**
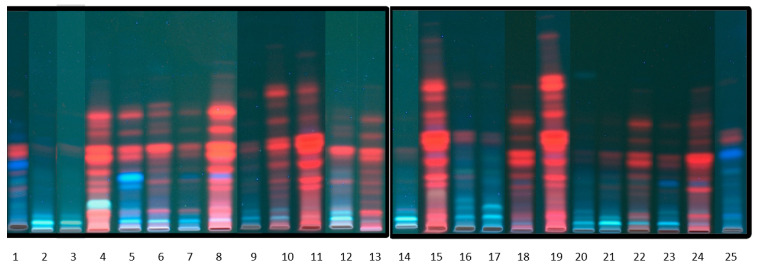
Example TLC chromatograms of radish–sprouts (1), radish–roots (2), radish–spots (3), radish–leaves (4), broccoli–sprouts (5), broccoli–flowers (6), Brussels sprouts–leaves (7), kale–leaves (8), canola–roots (9), canola–spots (10), canola–leaves (11), conehead cabbage–leaves (12), kohlrabi–spots (13), kohlrabi–edible spot (14), kohlrabi–leaves (15), green cabbage–leaves (16), Chinese cabbage–midvein (17), Chinese cabbage–leaves (18), cauliflower–leaves (19), cauliflower–head (20), cauliflower–midvein (21), Pak choi cabbage–leaves (22), red cabbage–leaves (23), Italian cabbage–leaves (24) and kale–sprouts (25).

**Figure 4 molecules-26-04734-f004:**
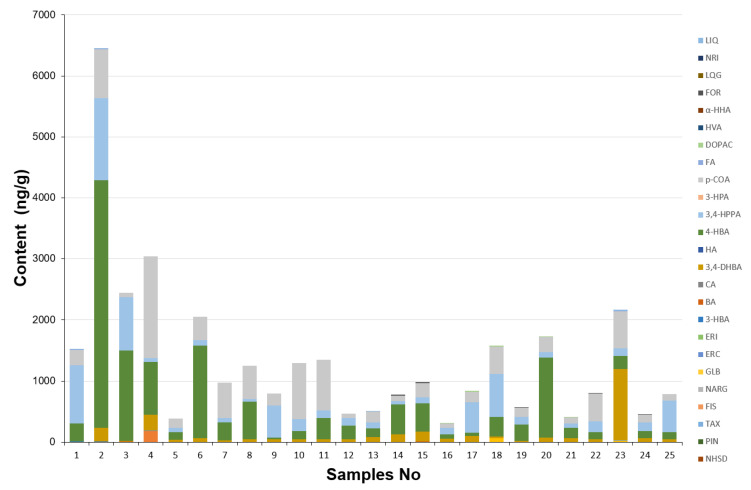
Content of phenolic acids and flavonoids determined in plant materials (*n* = 3) (the abbreviations of polyphenols in Section 2.4). 1—Radish–sprouts; 2—Radish–roots; 3—Radish–spots; 4—Radish–leaves; 5—Broccoli–sprouts; 6—Broccoli–flowers; 7—Brussels sprouts–leaves; 8—Kale–leaves; 9—Canola–roots; 10—Canola–spots; 11—Canola–leaves; 12—Conehead cabbage–leaves; 13—Kohlrabi–spots; 14—Kohlrabi–edible spot; 15—Kohlrabi–leaves; 16—Green cabbage–leaves; 17—Chinese cabbage–midvein; 18—Chinese cabbage–leaves; 19—Cauliflower–leaves; 20—Cauliflower–head; 21—Cauliflower–midvein; 22—Pak Choi cabbage–leaves; 23—Red cabbage–leaves; 24—Italian cabbage–leaves; and 25—Kale–sprouts.

**Figure 5 molecules-26-04734-f005:**
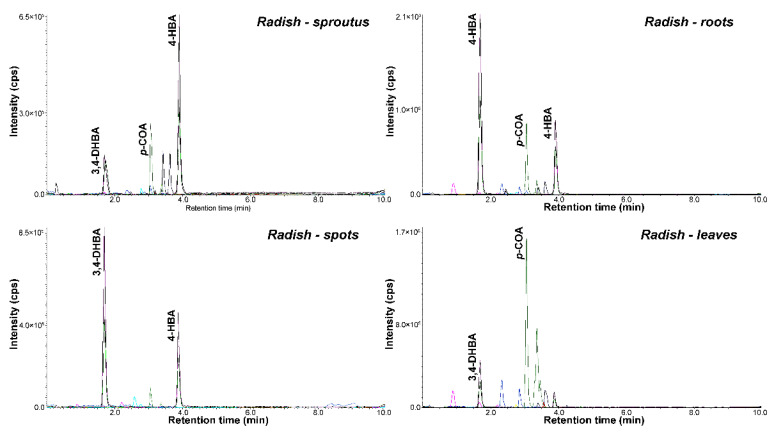
Representative chromatograms obtained for extracts of radish sprouts, radish roots, radish spots and radish leaves using the UHPLC-MS/MS method.

**Figure 6 molecules-26-04734-f006:**
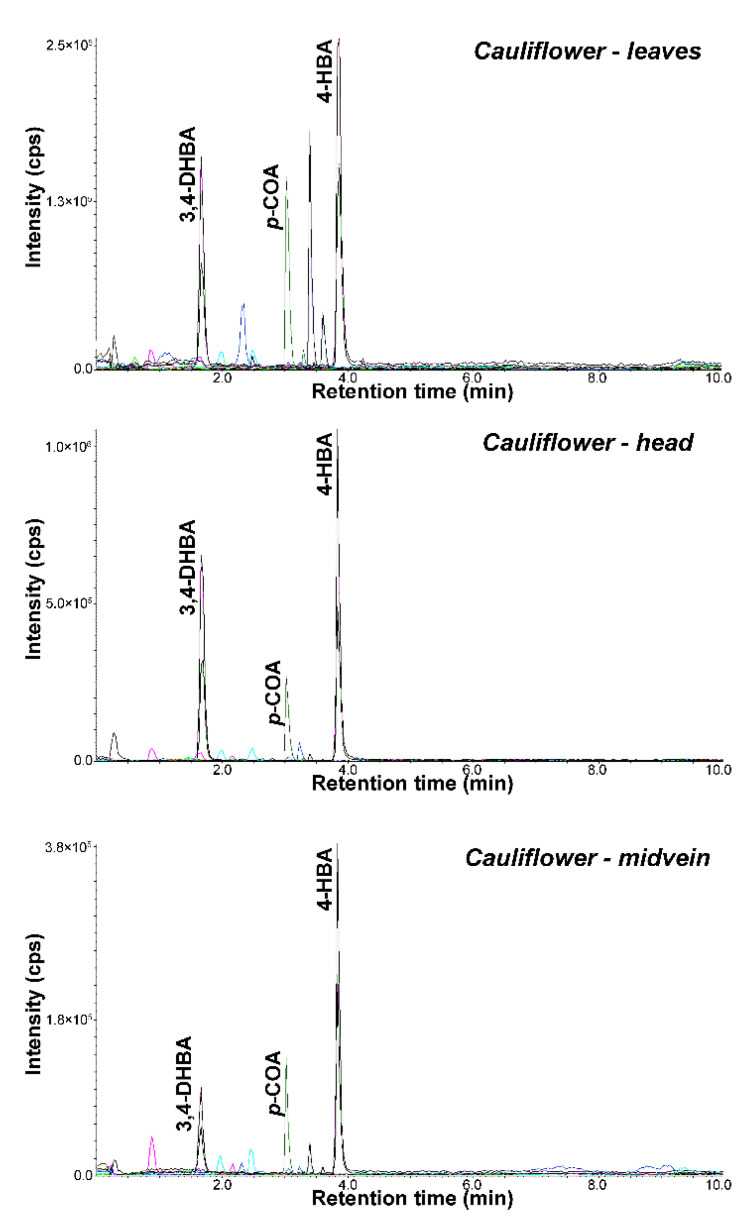
Representative chromatograms obtained for extracts of cauliflower leaves, cauliflower head and cauliflower midvein using the UHPLC-MS/MS method.

**Figure 7 molecules-26-04734-f007:**
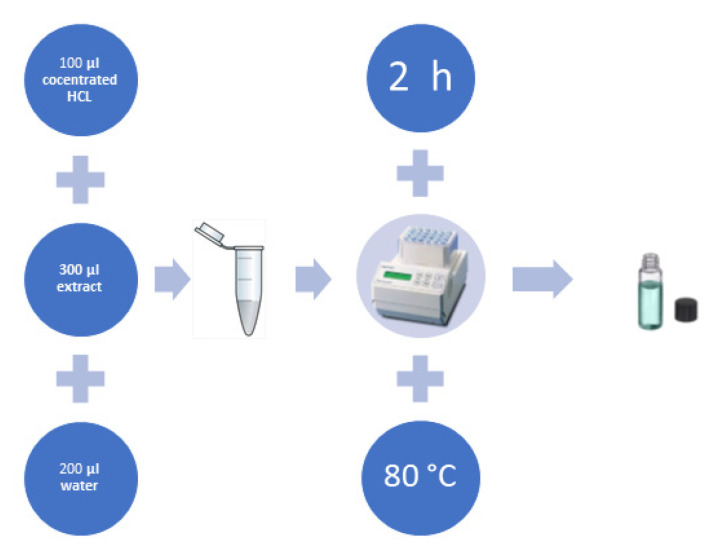
The schematics for acid hydrolysis of plant extracts.

**Table 1 molecules-26-04734-t001:** The extraction efficiency (*EY*) (*n* = 3).

No	Sample Type	The Extraction Yield (*EY*) [%] ± SD ^a^
1	Radish–sprouts	22.0 ± 2.11
2	Radish–roots	11.73 ± 1.18
3	Radish–spots	10.25 ± 1.04
4	Radish–leaves	6.22 ± 0.58
5	Broccoli–sprouts	19.96 ± 1.97
6	Broccoli–flowers	24.13 ± 2.42
7	Brussels sprouts–leaves	4.84 ± 0.35
8	Kale–leaves	16.86 ± 1.57
9	Canola–roots	8.00 ± 0.88
10	Canola–spots	11.88 ± 1.15
11	Canola–leaves	12.54 ± 1.19
12	Conehead cabbage–leaves	10.43 ± 1.02
13	Kohlrabi–spots	42.42 ± 4.15
14	Kohlrabi–edible spot	11.67 ± 1.09
15	Kohlrabi–leaves	14.23 ± 1.31
16	Green cabbage–leaves	16.06 ± 1.52
17	Chinese cabbage–midvein	23.22 ± 2.18
18	Chinese cabbage–leaves	13.59 ± 1.14
19	Cauliflower–leaves	7.27 ± 0.67
20	Cauliflower–head	18.73 ± 1.78
21	Cauliflower–midvein	28.75 ± 2.65
22	Pak Choi cabbage–leaves	17.49 ± 1.75
23	Red cabbage–leaves	15.10 ± 1.47
24	Italian cabbage–leaves	17.19 ± 1.57
25	Kale–sprouts	20.93 ± 2.13

^a^ SD–standard deviation.

**Table 2 molecules-26-04734-t002:** The total values of antioxidant activity and phenolic compounds content in extracts of *Brassicaceae* plants.

Sample No	%*RSA*	TEAC [mM]	GAE [mg/mL]	Total Phenolic Content [mg/g]
1	86.73 ± 0.67	81.86	0.85	8.47
2	40.09 ± 0.18	24.44	0.47	4.67
3	41.12 ± 0.19	26.43	0.51	4.86
4	54.55 ± 0.64	14.36	0.59	5.87
5	84.35 ± 0.64	78.45	0.80	8.03
6	41.97 ± 0.09	18.22	0.44	4.37
7	40.29 ± 1.00	15.84	0.44	4.40
8	52.54 ± 0.18	23.14	0.66	6.62
9	30.89 ± 0.55	9.19	0.35	3.47
10	42.88 ± 0.28	13.39	0.41	4.10
11	51.24 ± 0.69	21.04	0.45	4.52
12	38.02 ± 2.20	47.77	0.48	4.77
13	45.97 ± 1.23	10.65	0.36	3.55
14	20.77 ± 1.83	23.53	0.40	3.99
15	63.89 ± 1.19	32.23	0.56	5.60
16	76.59 ± 0.09	87.16	0.63	6.27
17	80.23 ± 0.46	72.39	0.63	6.26
18	67.19 ± 0.18	45.87	0.47	4.67
19	45.86 ± 0.46	28.58	0.51	5.13
20	61.55 ± 0.28	43.98	0.43	4.33
21	42.43 ± 2.75	49.28	0.45	4.46
22	29.98 ± 0.94	27.35	0.43	4.33
23	58.12 ± 0.92	48.90	0.64	6.44
24	56.89 ± 0.64	71.25	0.59	5.89
25	60.65 ± 0.09	94.73	0.77	7.72

**Table 3 molecules-26-04734-t003:** Validation data for determination of polyphenols.

Analyte	Linear Range (ng/g)	R^2^	*LOD*/*LOQ* (ng/g)	Nominal Concentration (ng/g)	Intra-Day		Inter-Day		Matrix Effect *ME* (%)
					CV ^a^ (%)	RE ^b^ (%)	CV ^a^ (%)	RE ^b^ (%)	
3,4-DHBA	0.4–500	0.9951	0.13/0.4	300	2.14	1.36	3.63	2.51	−4.58
				10	3.84	2.41	5.47	3.96	−6.98
α-HHA	0.4–500	0.9924	0.13/0.4	300	4.12	−2.15	6.18	−3.74	−8.47
				10	5.47	−3.47	7.56	−5.25	−7.36
DOPAC	0.4–500	0.9974	0.13/0.4	300	1.73	−4.28	3.58	−5.87	2.57
				10	3.69	−5.39	5.17	−6.39	4.65
4-HBA	0.4–500	0.9936	0.13/0.4	300	2.58	−2.76	3.81	−4.85	5.36
				10	4.69	−3.33	4.83	−6.97	6.17
CA	0.8–500	0.9947	0.26/0.8	300	1.93	2.15	4.51	4.28	−5.12
				10	2.17	4.87	6.37	5.78	−6.47
HA	0.4–500	0.9991	0.13/0.4	300	4.83	−1.79	7.42	−3.69	−2.14
				10	5.39	−3.65	8.69	−4.83	−6.07
3-HBA	0.4–500	0.9989	0.13/0.4	300	2.11	−0.94	4.41	−2.81	3.19
				10	4.78	−2.75	5.93	−5.12	5.47
3-HPA	0.4–500	0.9978	0.13/0.4	300	4.32	3.54	6.71	4.32	2.58
				10	6.71	4.39	8.54	6.96	4.96
HVA	0.8–500	0.9990	0.26/0.8	300	2.95	−3.27	4.33	−5.63	−3.96
				10	3.69	−5.83	7.05	−8.11	−5.87
3,4-HPPA	20–500	0.9963	6.67/20	300	0.87	−2.94	3.82	−4.75	−4.12
				50	2.47	−4.65	4.57	−6.35	−6.05
*p*-COA	0.4–500	0.9952	0.13/0.4	300	3.98	1.23	5.71	2.85	4.78
				10	4.72	3.67	6.87	5.78	5.39
FA	0.4–500	0.9932	0.13/0.4	300	2.56	2.26	4.41	4.59	−1.47
				10	4.78	4.63	5.89	6.32	−4.36
ERC	0.4–500	0.9981	0.13/0.4	300	3.66	−3.15	4.87	−4.59	3.97
				10	5.21	−3.98	6.39	−5.93	5.82
LIQ	0.4–500	0.9994	0.13/0.4	300	4.17	−4.71	5.71	−6.47	2.74
				10	6.41	−5.02	8.53	−7.89	4.69
RUT	0.4–500	0.9993	0.13/0.4	300	3.25	2.47	4.52	4.52	3.55
				10	4.17	3.69	5.36	6.36	5.41
TAX	0.4–500	0.9947	0.13/0.4	300	2.87	4.52	4.23	5.29	2.69
				10	4.63	5.87	6.87	7.85	3.74
BA	0.8–500	0.9968	0.26/0.8	300	1.85	−2.78	2.54	−3.89	4.11
				10	3.64	−5.69	4.85	−7.25	5.07
NRI	0.8–500	0.9920	0.26/0.8	300	2.87	−3.21	4.41	−4.25	−6.87
				10	3.54	−4.87	6.52	−6.78	−8.15
NARG	0.8–500	0.9994	0.26/0.8	300	0.47	−5.70	1.23	−6.23	4.25
				10	3.69	−6.42	4.32	−8.52	5.78
HSD	0.4–500	0.9987	0.13/0.4	300	4.21	2.14	6.05	3.46	5.23
				10	5.87	4.78	7.93	4.85	6.90
NHSD	0.4–500	0.9965	0.13/0.4	300	3.78	3.58	5.25	5.39	2.14
				10	4.93	4.96	6.38	6.87	4.79
FIS	0.4–500	0.9971	0.13/0.4	300	2.41	1.64	4.21	2.58	−3.67
				10	5.87	2.83	7.28	4.65	−4.52
LQG	0.4–500	0.9939	0.13/0.4	300	4.28	−5.22	6.48	−6.87	−6.21
				10	6.14	−5.89	8.52	−7.41	−7.56
ERI	0.4–500	0.9914	0.13/0.4	300	5.02	1.74	6.35	2.69	−5.47
				10	5.89	3.69	6.98	4.78	−8.73
QUE	0.8–500	0.9975	0.26/0.8	300	2.73	4.14	4.11	6.15	−5.21
				10	3.85	5.63	5.28	7.25	−6.78
NAR	0.4–500	0.9993	0.13/0.4	300	1.73	2.54	2.55	4.32	1.63
				10	4.62	4.36	5.87	5.69	2.87
HST	0.8–500	0.9945	0.26/0.8	300	3.98	−5.32	3.69	−6.14	3.69
				10	5.87	−6.12	7.21	−8.25	5.74
FOR	0.4–500	0.9968	0.13/0.4	300	2.02	−1.56	4.20	−3.69	2.46
				10	4.96	−3.54	5.93	−4.75	6.89
PIN	0.4–500	0.9972	0.13/0.4	300	1.52	2.47	2.47	3.62	1.25
				10	3.69	3.69	4.89	5.55	4.98
GLB	0.4–500	0.9954	0.13/0.4	300	0.57	4.52	1.47	5.87	−3.57
				10	3.58	5.69	5.63	6.45	−4.82

^a^ RSD relative standard deviation. ^b^ CV coefficient of variation.

**Table 4 molecules-26-04734-t004:** List of investigated samples.

No	Sample Type	Latin Name
1	Radish–sprouts	*Raphanus sativus* var. Sativus
2	Radish–roots	*Raphanus sativus* var. Sativus
3	Radish–spots	*Raphanus sativus* var. Sativus
4	Radish–leaves	*Raphanus sativus* var. Sativus
5	Broccoli–sprouts	*Brassica oleracea* L. var. italica Plenck
6	Broccoli–flowers	*Brassica oleracea* L. var. italica Plenck
7	Brussels sprouts–leaves	*Brassica oleracea* L. var. gemmifera (DC.) Zenker
8	Kale–leaves	*Brassica oleracea* L. var. sabellica L.
9	Canola–roots	*Brassica napus* L.
10	Canola–spots	*Brassica napus* L.
11	Canola–leaves	*Brassica napus* L.
12	Conehead cabbage–leaves	*Brassica oleracea* ‘Cour di Bue Grosso’
13	Kohlrabi–spots	*Brassica oleracea* var. gongylodes L.
14	Kohlrabi–edible spot	*Brassica oleracea* var. gongylodes L.
15	Kohlrabi–leaves	*Brassica oleracea* var. gongylodes L.
16	Green cabbage–leaves	*Brassica oleracea* L. var. capitata L.
17	Chinese cabbage–midvein	*Brassica rapa* L. subsp. pekinensis
18	Chinese cabbage–leaves	*Brassica rapa* L. subsp. pekinensis
19	Cauliflower–leaves	*Brassica oleracea* L. var. botrytis L.
20	Cauliflower–head	*Brassica oleracea* L. var. botrytis L.
21	Cauliflower–midvein	*Brassica oleracea* L. var.botrytis L.
22	Pak Choi cabbage–leaves	*Brassica rapa* L. subsp. chinensis Hanelt
23	Red cabbage–leaves	*Brassica oleracea* var. capitata f. rubra
24	Italian cabbage–leaves	*Brassica oleracea* L. var. sabauda L.
25	Kale–sprouts	*Brassica oleracea* L. var. sabellica L.

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
