# Peer review of "Separation and Determination of Chemopreventive Phytochemicals of Flavonoids from *Brassicaceae* Plants"

_molecules, 2021, doi:10.3390/molecules26164734_

Round 1
Reviewer 1 Report
Thank you for submitting the manuscript “Separation and determination of chemopreventive phytochemicals of flavonoids from Brassicaceae plants” to Molecules. The work is well written, the experiment was very well designed and conducted, and the results are encouraging.
So, I even have few suggestions for this manuscript:
1) Lines 33 and 36. There is an inconsistency in the use of the comma. My suggestion is:
Item A, Item B, or/and Item C
OR
Item A, Item B or/and Item C
Please correct all text.
2) Please correct figure 1 as the caption is underlined for no reason.
3) Line 85: I believe the name of this subsection is material and methods.
4) Figures should be self-explanatory. In Figure 6, in addition to including the number of samples, it is necessary to provide a legend regarding the abbreviations used. Apply to all figures.
Author Response
Dear Editor(s) and Reviewers of Molecules
Thank you for your review of our manuscript.
We have corrected the manuscript according to your comments. All the changes made are indicated using the red font and the yellow highlighter in the corrected version of the manuscript. In reply to your remarks, we would cite the remarks of the reviewers and respond to them one by one. We would like to express our gratitude for the comments on this manuscript, which we believe greatly improved its clarity and overall quality.
The responses to the reviews are attached.

Reviewer 2 Report
This work presents an interesting characterisation of polyphenols from various part of plants belonging to Brassicaceae family through thin-layer chromatography and ultra-performance liquid chromatography coupled to mass spectrometer.
The results show that the plants have a high content of polyphenols and therefore a good antioxidant activity and health-promoting benefits. For this purpose, the work could be accepted for publication in Molecules, with some modifications, as suggest below:
- Miss the references in the introduction and also in lines 26, 28, 30, 33, 35, please add the specific reference for each sentence.
- In 41 line should be added a last reference such as “Rapid screening and characterisation of glucosinolates in 25 Brassicaceae tissues by UHPLC-Q-exactive orbitrap-MS” (Food Chemistry 2021, 365, 130493) and “Determination of the metabolite content of Brassica juncea cultivars using comprehensive two-dimensional liquid chromatography coupled with a photodiode array and mass spectrometry detection.” (Molecules 25, no. 5: 1235.)
- The aim of the study should be more detailed in the introduction section. Please changes it.
- In line 87, change “were” with “are”
- Line 90, use “grounded” instead of “ground”.
- Please, in the table 1 “Raphanus sativus” should be in italic
- In section 2.2, should be added a reference for the extraction method used.
- Line 96 “Static extraction” is referred for the maceration? Please replace the term.
- Line 100, please add the size of the syringe filters.
- Please cite the procedure of hydrolysis used.
- Please add the name of the standards in line 112.
- How many milligrams of standard compounds were dissolved in 1 mL of methanol? please reports it in line 114.
- Please, detail the 10 levels of calibration curve in line 116.
- “0.4 to 500 ng/g” is a very low concentration please can you confirmed this units?
- Please add the column used in the section 2.4
- Please details the acronym “HPTLC” in line 127.
- Line 186: Please report the gradient program used.
- Please, can you report the MRM transitions for each compound in the text or like table in the supplementary materials.
- Line 193: “transitions were” instead “transition was”
- Please report the guidelines selected for the Validation method in the line 200.
- Please correct the equation 1: LOQ= 3LOQ? and report the reference.
- Please add the reference in the 2 equation.
- Are you sure the equation 3 is referred at extraction efficiency? Probably is referred at extraction yield.
- How many extractions have you performed? Please report in the table 2.
- Line 285, please you should add more reference.
- Please add the tree digits after the decimal point in the 0.99 value.
- How were obtain the value of 527 ng/g of 3,4 HPPA; 924 and 833 ng/g of p-COA, 1167 of 3,4-DHBA, if the maximum concentration of the standard compounds were 500 ng/g?
- The authors can briefly discuss because some flavonoids were not quantified? Line 361.
- The resolution of the Figures 6 and 7 are very poor, please improve it and for each peak report the name of the compound.
- Please, for a better understanding of the qualitative profile for each sample you should add a new table that report all compounds identified in the all samples.
- Line 32, 74, 287, 295 “e.g.”,185 “v/v”, 205 “vs.”in italic.
Author Response

(The authors gave the same response as above.)

Round 2
Reviewer 2 Report
The manuscript can be accepted in this form, the authors have done all revisions request.